# Identification of TALE Transcription Factor Family and Expression Patterns Related to Fruit Chloroplast Development in Tomato (*Solanum lycopersicum* L.)

**DOI:** 10.3390/ijms23094507

**Published:** 2022-04-19

**Authors:** Jin Wang, Pan Zhao, Baohui Cheng, Yanhong Zhang, Yuanbo Shen, Xinyu Wang, Qinghua Zhang, Qianqi Lou, Shijie Zhang, Bo Wang, Shiming Qi, Yushun Li, Md. Monirul Islam, Tayeb Muhammad, Fei Zhang, Yan Liang

**Affiliations:** 1College of Horticulture, Northwest A&F University, Yangling, Xianyang 712100, China; jw6127@nwafu.edu.cn (J.W.); zhaopan2019@nwafu.edu.cn (P.Z.); 2021055286cbh@nwafu.edu.cn (B.C.); 2021055322@nwafu.edu.cn (Y.Z.); syb777@nwafu.edu.cn (Y.S.); wyx0802@nwafu.edu.cn (X.W.); zhangqh2020055352@nwafu.edu.cn (Q.Z.); qqlou1996@nwafu.edu.cn (Q.L.); 13289316916zsj@nwafu.edu.cn (S.Z.); wbd@nwafu.edu.cn (B.W.); qishiming2008@nwafu.edu.cn (S.Q.); liyushun2016@nwafu.edu.cn (Y.L.); monirul@nwafu.edu.cn (M.M.I.); tayebmuhammad@nwsuaf.edu.cn (T.M.); feizhang@nwsuaf.edu.cn (F.Z.); 2Directorate of Agriculture Extension, Merged Areas, Peshawar 25000, Pakistan

**Keywords:** TALE, tomato, chloroplast development, fruit development, regulatory function

## Abstract

The *TALE* gene family is an important transcription factor family that regulates meristem formation, organ morphogenesis, signal transduction, and fruit development. A total of 24 genes of the TALE family were identified and analyzed in tomato. The 24 SlTALE family members could be classified into five BELL subfamilies and four KNOX subfamilies. *SlTALE* genes were unevenly distributed on every tomato chromosome, lacked syntenic gene pairs, and had conserved structures but diverse regulatory functions. Promoter activity analysis showed that cis-elements responsive to light, phytohormone, developmental regulation, and environmental stress were enriched in the promoter of *SlTALE* genes, and the light response elements were the most abundant. An abundance of TF binding sites was also enriched in the promoter of *SlTALE* genes. Phenotype identification revealed that the *green shoulder* (*GS*) mutant fruits showed significantly enhanced chloroplast development and chlorophyll accumulation, and a significant increase of chlorophyll fluorescence parameters in the fruit shoulder region. Analysis of gene expression patterns indicated that six *SlTALE* genes were highly expressed in the *GS* fruit shoulder region, and four *SlTALE* genes were highly expressed in the parts with less-developed chloroplasts. The protein-protein interaction networks predicted interaction combinations among these *SlTALE* genes, especially between the BELL subfamilies and the KNOX subfamilies, indicating a complex regulatory network of these *SlTALE* genes in chloroplast development and green fruit shoulder formation. In conclusion, our result provides detailed knowledge of the *SlTALE* gene for functional research and the utilization of the *TALE* gene family in fruit quality improvement.

## 1. Introduction

Homeobox (Homologous alien box) genes, which produce the same phenotype when mutated, are a family of transcription factors (TFs) that are highly conserved in plants, animals, and fungi [1]. The first homeobox gene was identified in *Drosophila*, and *KNOTTED1* (*KN1* or *ZmKN1*), obtained from a functionally acquired mutant of maize, was the first homeobox gene isolated from plants [2,3]. Homeobox genes encode a typical DNA-binding domain of 60 amino acids, known as the homeodomain (HD), which forms three-helix regions; the first and second helices form a loop structure, and the second and third helices form a helix-corner-helical structure [4]. Many homeobox genes have been subsequently identified from all major eukaryotic lineages [5]. Plant homeobox genes can be classified into 11 classes based on their structural characteristics and phylogeny: HD-ZIP (including HD-ZIP I to HD-ZIP IV), WOX, NDX, PHD, PLINC, LD, DDT, SAWADEE, PINTOX, KNOX, and BELL [6].

Unlike typical homeobox genes, the *TALE* (Three Amino acid Loop Extension) homeobox genes encode an atypical structure of 63 amino acids with three additional amino acid residues (P-Y-P) inserted between the first and the second helix [7,8,9]. The TALE family consists of five members, but only members of the KNOX (KNOTTED-like homeodomain) subfamilies and BELL (BEL1-like homeodomain) subfamilies have been observed in plants. KNOX proteins contain a MEINOX (from MEIS and KNOX) domain composed of KNOX1 and KNOX2 subdomains separated by a flexible linker, an ELK domain, and a TALE homeodomain. The MEINOX domain mediates the interaction between KNOX and BELL proteins [10,11]. In addition, KNATM, a novel TALE family member without a homeodomain, interacts with BELL proteins through the MEINOX domain and interacts with BP via the BPID (BP interacting domain) [11,12]. BELL proteins contain a TALE homeodomain and a MID (MEINOX interacting domain, or POX) composed of the SKY and BELL domains [13,14]. The interaction of BELL proteins with KNOX proteins targets the KNOX-BELL heterodimer to the nucleus [15,16]. The function of the ELK domain remains unclear, and it might be related to interactions between proteins. However, some studies have suggested that it contains a nuclear localization signal and might be involved in inhibiting the transcription of genes [14].

TALE family members play an important role in hormone regulation, signal transduction, meristem formation, organ morphogenesis, and the development of plants [17,18,19,20,21,22]. For example, ATH1 (Arabidopsis homeobox 1) interacts with STM (homeobox protein SHOOT MERISTEMLESS) and KNAT2 (Arabidopsis KNOX 2) and plays a role in the development of meristems and inflorescence tissue [23,24]. The interaction between BLH6 and KNAT7 affects the formation of the secondary cell wall [25]. *TKN4* and *TKN2* regulate chloroplast development and the distribution of chloroplasts in tomato fruit by targeting *APRR2-like* and *GLK2* [22,26]. *SlBEL11* is responsible for regulating chloroplast development, chlorophyll accumulation, and the chlorophyll quality of tomato fruit by modifying the expression of *POR*, *CAB*, and *TKN2* [27]. *SlBEL4* regulates chloroplast development and cell wall metabolism via the transcriptional regulation of *CHLD*, *POR*, *TKN2*, and *PE* [17]. Although TALE family members involved in the regulation of plant development have been well characterized, the functions of *TALE* genes in tomato fruits have been less studied by comparison because TALE family members have diverse functions and often show complex interactions with other genes.

To identify potential TALE members and understand the evolution and function of *TALE* genes in tomato, genome-wide identification and analysis of the *TALE* genes were conducted. The expression of *TALE* genes was evaluated between the stem end (fruit shoulder area) and the base (fruit navel area) of non-uniform ripening tomato fruit. The results of the systematic identification and analysis of *TALE* genes provide valuable insights into the regulatory mechanisms of *TALE* genes in chloroplast development and the formation of green fruit shoulders during tomato fruit development.

## 2. Results

### 2.1. Identification and Sequence Analysis of SlTALE Family Members

A total of 52 and 28 candidate TALE members were identified from the tomato genome by HMMER search and BLASTP, respectively. Next, the two identification results were combined and taken together. The results were further verified by PfamScan and SMART; a total of 24 *SlTALE* members were confirmed in tomato (Table 1). Fourteen *SlTALE* members belonged to the BELL subfamilies and possessed a POX domain and homeodomain, and the remaining 10 members belonged to the KNOX subfamilies. In the KNOX subfamily, six members contained KNOX1, KNOX2, ELK, and HOX domains, but the KNOX2 domain of SlTALE2 was located at the same position as the first ELK. SlTALE12 and SlTALE18 lacked a KNOX1 domain and ELK domain, respectively, and SlTALE7 and SlTALE13 lacked both ELK and HOX domains (Appendix A).

The 24 *SlTALE* genes were predicted to encode proteins ranging from 169 to 726 amino acids in length. The molecular weights ranged from 19.5 to 79.2 kDa, and isoelectric points ranged from 4.38 to 8.91. The instability index ranged from 42.9 to 65.2, and the aliphatic index varied from 60.7 to 86.4, suggesting that SlTALEs were all unstable and fat-soluble proteins. The GRAVY value was between −0.995 and −0.385, suggesting that SlTALEs were all hydrophilic proteins. The signal peptide prediction showed that all SlTALE proteins were non-secreted proteins. Subcellular localization prediction suggested that all SlTALE proteins were distributed in the nucleus (Table 1).
ijms-23-04507-t001_Table 1Table 1Information of SlTALE family members identified in the tomato genome.Gene NameGene IDKnown AsIntronsAAMW (kDa)IPIIAIGRAVYSPPSL*SlTALE1*Solyc01g007070
370037.916.2252.1866.21−0.620.0017NC*SlTALE2*Solyc01g100510TKN4 433571.885.3047.0166.75−0.710.0030NC*SlTALE3*Solyc01g109980
364569.745.1253.7661.89−0.740.0008NC*SlTALE4*Solyc02g065490
362439.656.4745.1163.75−0.730.0006NC*SlTALE5*Solyc02g081120LeT6, TKN2 [28]335479.195.7245.6160.68−0.700.0131NC*SlTALE6*Solyc02g089940BIP [29]372620.696.7247.6361.76−0.590.0007NC*SlTALE7*Solyc03g112740
318440.094.3844.9586.41−0.390.0004NC*SlTALE8*Solyc04g077210TKN1 [30]435570.565.8645.0061.61−0.820.0037NC*SlTALE9*Solyc04g079830
662745.586.9643.6872.98−0.700.0010NC*SlTALE10*Solyc04g080780BL1, BEL11 [27]340073.896.0748.1877.55−0.490.0007NC*SlTALE11*Solyc04g080790
366119.875.7144.5672.27−0.620.0006NC*SlTALE12*Solyc05g005090
416919.505.3550.6363.55−0.970.0004NC*SlTALE13*Solyc06g072480KD1, TKD1 [31]317177.665.2446.3869.59−0.670.0004NC*SlTALE14*Solyc06g074120BL2 [32]369948.016.8652.6269.77−0.710.0038NC*SlTALE15*Solyc07g007120LeT12, THox1 [28]543139.645.6761.9361.60−0.940.0004NC*SlTALE16*Solyc08g041820
434962.665.3760.4677.97−0.750.0006NC*SlTALE17*Solyc08g065420BL4, BEL4 [17]355935.306.3248.1177.12−0.420.0009NC*SlTALE18*Solyc08g080120THox2 [32]431072.656.0665.1679.90−0.720.0018NC*SlTALE19*Solyc08g081400
364455.976.2251.2968.12−0.760.0016NC*SlTALE20*Solyc09g011380
350430.517.6942.8971.17−0.600.0004NC*SlTALE21*Solyc10g086640
526872.928.9144.0680.04−0.530.0003NC*SlTALE22*Solyc11g068950
466175.786.8444.7972.12−0.660.0159NC*SlTALE23*Solyc11g069890
368639.276.7253.9174.71−0.500.0020NC*SlTALE24*Solyc12g010410
534037.915.5653.1066.50−0.990.0003NCNote: AA—amino acid sequence length, MW—molecular weight, IP—isoelectric point, II—Instability Index, AI—Aliphatic Index, GRAVY—grand average of hydropathicity, SP—signal peptide, Introns—intron number, PSL—protein subcellular localization; NC—Nucleus.

### 2.2. Phylogenetic Analysis of the TALE Family

A phylogenetic tree of *TALE* family genes from various plant species was constructed to characterize the evolutionary relationships among these genes, elucidate their biological functions, and clarify the similarities in the SlTALE proteins among tomato and other plants (Figure 1, Appendix A).

The 24 SlTALE family members could be classified into the BELL subfamilies and KNOX subfamilies. The BELL subfamilies contained five groups: BELL-I (SlTALE17), BELL-II (SlTALE20, SlTALE21, and SlTALE23), BELL-III (SlTALE4, SlTALE6, SlTALE9, and SlTALE19), BELL-IV (SlTALE1, SlTALE14, and SlTALE22), and BELL-V (SlTALE3, SlTALE10, and SlTALE11), and the KNOX subfamilies contained four groups: KNOX-I (SlTALE7 and SlTALE13), KNOX-II (SlTALE15, SlTALE16, SlTALE18, and SlTALE24), KNOX-III (SlTALE2 and SlTALE12), and KNOX-IV (SlTALE5 and SlTALE18) (Figure 1).

### 2.3. Analysis of Gene Structures, Chromosomal Locations, and Conserved Motifs

*SlTALE* genes were mapped to the tomato genome to study their distribution. *SlTALE* genes were randomly and non-randomly distributed across all 12 chromosomes; there were 2–4 members on all chromosomes except 3, 5, 7, 9, 10, and 12, which had only 1 member (Figure 2A). Both chromosomes 4 and 8 had the highest (16.66%) number of genes compared with the other chromosomes. Furthermore, most of the *SlTALE* genes were located at both ends of tomato chromosomes, especially at the long arm. The syntenic relationship analysis showed that there were no collinear pairs of *SlTALE* genes, but tandem duplications were observed between *SlTALE10* and *SlTALE11* as well as between *SlTALE22* and *SlTALE23* (Figure 2A).

Structural analysis showed that the number of exons and introns of the *SlTALE* genes ranged from 4 to 7 and from 3 to 6, respectively (Figure 2D). The gene structure and distribution of conserved motifs were similar among members of the same subfamily. For example, four KNOX-II members (*SlTALE15*, *SlTALE16*, *SlTALE18*, and *SlTALE24*) contained five to six exons and four identical motifs, and three BELL-IV members (*SlTALE1*, *SlTALE14*, and *SlTALE22*) contained five exons and seven identical motifs (Figure 2C). The above results indicated that the *SlTALE* family members were conserved, and the differences in gene structure or protein-conserved motifs might be related to their various biological functions.

### 2.4. Analysis of Cis-Elements and TF Binding Sites

To characterize the functions and regulatory mechanisms of *SlTALE* family genes, the *cis*-elements and TF binding sites of their promoter regions were analyzed (Figure 3, Appendix A). A total of 41 different cis-elements identified in the promoter of *SlTALE* family genes were detected in response to 16 signal factors: light, phytohormone (abscisic acid, auxin, gibberellin, and zein), developmental regulation (circadian control, endosperm, flavonoid biosynthesis, endosperm, and meristem), and environmental stress (anaerobic, anoxic, defense and stress, drought, low temperature, methyl jasmonate, and salicylic acid). A total of 21 *cis*-elements involved in light responsiveness regulation were detected, and they accounted for 53.4% of all *cis*-elements (Figure 3A,B, Appendix A). According to the TF binding site analysis, a total of 1045 transcription factor binding sites belonging to 33 different TFs were identified in the promoters of *SlTALE* family genes. (Figure 3C, Appendix A). In plants, these TFs regulate various metabolic pathways, suggesting that the regulatory functions of *SlTALE* family genes are diverse.

### 2.5. Interaction Network of the SlTALE Proteins

Clarifying the functional relationships among SlTALE proteins is important for characterizing their potential functions and regulatory pathways. A protein–protein interaction network among SlTALE proteins was predicted using STRING software (Figure 4). Most of the SlTALE proteins could interact with multiple other members except SlTALE2 and SlTALE10, as no interactions of SlTALE2 and SlTALE10 with other members were observed. SlTALE11 and BELL-III subfamily members (SlTALE4, SlTALE6, SlTALE9, and SlTALE19) were central to the interaction network and interacted strongly with most of the KNOX subfamily members, including KNOX-IV (SlTALE5 and SlTALE8), 3 KNOX-II members (SlTALE15, SlTALE16, and SlTALE24), and 1 KNOX-III member (SlTALE12). In addition, BELL-II subfamily members (SlTALE20, SlTALE21, and SlTALE23) might interact with KNOX-IV subfamily members (SlTALE5 and SlTALE8) as well as 1 KNOX-III member (SlTALE12). BELL-IV subfamily members (SlTALE1, SlTALE14, and SlTALE22) might interact with SlTALE18 and SlTALE18. Moreover, the KNOX-I member SlTALE7 might interact with BELL-II (SlTALE20, SlTALE21, and SlTALE23), BELL-III (SlTALE4, SlTALE6, SlTALE9, and SlTALE19), and KNOX-II (SlTALE15, SlTALE16, SlTALE18, and SlTALE24). Functional interactions, especially gene co-occurrence and co-expression, were mainly observed between the BELL subfamilies and the KNOX subfamilies, indicating that these genes might be synergistically involved in certain regulatory pathways.

### 2.6. Characterization of Fruit Phenotype and Chloroplast Development of the GS Mutant and WT

Unlike uniformly pigmented WT tomato fruit (TTD302A), the *GS* mutant was characterized by fruit with various pigments and a dark green fruit shoulder (Figure 5A). The shoulder of *GS* fruit accumulated significantly large amounts of chlorophyll compared with the WT (both in the shoulder and base) (Figure 5D). The photosynthetic performance of WT and *GS* fruits was assessed by determining the chlorophyll fluorescence parameters (Figure 5B,E). Chlorophyll fluorescence parameters, including minimal fluorescence (Fo), maximal fluorescence (Fm), and variable fluorescence (Fv), were significantly higher in *GS* mutant fruits than in WT fruits (Figure 5E). In addition, chloroplast development was accelerated and chloroplast number and size, the number of grana, and thylakoid density were higher in the shoulder of *GS* fruit (Figure 5C). These results suggested that the green color formation of the shoulder in *GS* mutant fruit was promoted by chloroplast development and chlorophyll biosynthesis and accumulation.

### 2.7. Conserved and Divergent Patterns of SlTALE Genes Expression during Chloroplast Development and Green Shoulder Formation in Tomato Fruit

The expression of genes is often related to their function. To gain insight into the characteristics and functions of *SlTALE* genes in tomato fruit, the expression patterns of genes were analyzed based on RNA-seq data (Appendix A). Most *SlTALE* genes were expressed during different fruit developmental stages in the *GS* mutant and WT, except for eight genes (*SlTALE4*, *SlTALE7*, *SlTALE8*, *SlTALE10*, *SlTALE12*, *SlTALE13*, *SlTALE17*, and *SlTALE23*) (Figure 6). Among the 16 expressed genes, 5 genes (*SlTALE1*, *SlTALE14*, *SlTALE15*, *SlTALE18*, and *SlTALE22*) had high expression levels but nonsignificant differences between the fruit shoulder of *GS* and WT. *SlTALE2* and *SlTALE5* exhibited similar expression patterns with significantly higher expression in the shoulder of *GS* compared with that of WT. The expression of *SlTALE16* was higher in *GS* than in WT at 28 dpa and P. *SlTALE24* was expressed exclusively in the fruit shoulder of WT before 28 dpa and was highly expressed in both *GS* and WT after 28 dpa. The expression profiles of *SlTALE20* and *SlTALE21* were similar with higher expression in *GS* than in WT. *SlTALE6*, *SlTALE9*, and *SlTALE19* (all of which are in the same subfamily) were more highly expressed in WT than in *GS*. *SlTALE3* and *SlTALE11* were more highly expressed in *GS* than in WT.

RNA-seq data of “Ailsa Craig” tomato (Appendix A) were used to investigate the expression profile of the *SlTALE* family genes in different tomato materials. *SlTALE* family genes had similar expression profiles in fruits from different tomato materials (Figure 6A,B). BELL-IV subfamily genes (*SlTALE1*, *SlTALE14*, and *SlTALE22*) and KNOX-II (*SlTALE15*, *SlTALE16*, *SlTALE18*, and *SlTALE24*) exhibited similar expression patterns in different tomato materials and different parts of the fruit, indicating that the functions of these genes were conserved but not necessarily involved in the regulation of chloroplast development in tomato fruit (Figure 6A,B).

### 2.8. Expression Analysis of SlTALE Genes in Tomato Fruit

To further investigate the *SlTALE* genes involved in chloroplast development and green shoulder formation in tomato fruit, the RNA-seq results of 16 expressed genes were validated via RT-qPCR (Figure 7 and Appendix A). These genes can be divided into three categories according to their expression levels in the shoulder and base of tomato fruit. Eight of the sixteen analyzed genes, *SlTALE2*, *SlTALE3*, *SlTALE5*, *SlTALE6*, *SlTALE11*, *SlTALE19*, *SlTALE20*, and *SlTALE24*, showed high expression in the *GS* fruit shoulder region (Gs) in more developed chloroplasts. Four analyzed genes, *SlTALE9*, *SlTALE18*, *SlTALE21*, and *SlTALE22*, showed high expression in the parts with less-developed chloroplasts such as Gb, Ws, and Wb. Four genes, *SlTALE1*, *SlTALE14*, *SlTALE15*, and *SlTALE16*, showed no difference in expression among all parts of the fruit (Figure 7). In general, the expression patterns of these 16 genes in tomato fruits were consistent with the RNA-seq data.

## 3. Discussion

TALE family genes play a crucial role in plant development and are widely distributed in the plant kingdom, but the number of these genes varies greatly among different plant species [33,34,35,36]. Tomato is one of the most widely grown vegetable crops, and it is an important model plant for studying fruit development [37]. Previous studies have shown that several TALE family genes regulate tomato fruit development, including fruit cell wall degradation, chloroplast development, pigment metabolism, fruit coloration and quality [17,22,27]. However, the number of *TALE* genes, their biological features, and regulatory functions in tomato fruit development remain unknown. In this study, 24 *SlTALE* family members were identified in tomato. Their biological features, structure, phylogenetic relationships, and expression regulation were characterized during tomato fruit development.

Gene organization, also known as exon-intron structure, plays a key role in the evolution of multigene families and can provide additional information for phylogenetic analysis [38,39]. Our results revealed a significant correlation between phylogeny and gene organization among *SlTALE*s, especially within the same groups and subfamilies (Figure 2). In addition, intron number is usually associated with the sensitivity of gene transcriptional regulation [40]. The number and length of introns are negatively correlated with the ability of plants to adapt to diverse developmental processes and environmental conditions [41,42]. The diverse gene structures and intron phases of *SlTALE*s in different subfamilies might be related to their distinct biological functions. There was also some similarity observed within groups and subfamilies, as these members shared similar domains and possessed a similar motif distribution (Figure 2). The consistency in gene organization, domains, motifs, and phylogenetic relationships demonstrates that these genes were correctly classified.

Gene duplications and tandem duplication of chromosomal regions affect the formation of gene families [43]. Chromosomal distribution and syntenic relationship analysis of *SlTALE* were performed to clarify the origin of the *SlTALE* family. The *SlTALE* genes were unevenly distributed on every tomato chromosome, and more *SlTALE* genes were located at the long arm of the chromosome, which facilitates the exchange of DNA fragments and gene recombination [44]. No collinear pairs of *SlTALE* genes were detected, and two tandem duplications were observed (Figure 2); this indicated that *SlTALE* family members might originate from different ancestors with high evolutionary diversity. These properties may explain the functional differences among *SlTALE* genes.

Promoter activity plays an important role in the regulation of gene functions [45]. Analysis results of the cis-elements and TF binding sites of the promoter regions revealed that *SlTALE* genes were found in response to various signal factors, such as light, phytohormone, developmental regulation, and environmental stress, and also are regulated by a variety of transcription factors (Figure 3 and Appendix A). These properties might also account for the diversity of the regulatory functions of *SlTALE* family members and indicate that *SlTALE* family genes play an important regulatory role in different biological processes of plant development, especially those related to light regulation (e.g., photosynthesis, plastid development, and pigment metabolism) [8,12].

The expression patterns of genes are related to their potential functions. We analyzed the expression profiles of the 24 *SlTALE* genes. Some *SlTALE* genes showed specific expression patterns between the shoulder and base of tomato fruit during fruit development (chloroplast development and green shoulder formation). Six *SlTALE* genes (*SlTALE2*, *SlTALE3*, *SlTALE5*, *SlTALE11*, *SlTALE20*, and *SlTALE21*) were highly expressed in the *GS* fruit shoulder region (Gs), and four *SlTALE* genes (*SlTALE6*, *SlTALE9*, *SlTALE19*, and *SlTALE24*) were highly expressed in the parts with less-developed chloroplasts, such as Gb, Ws, and Wb (Figure 6). These *SlTALE* genes, which show a latitudinal expression gradient, might regulate the establishment of a chloroplast developmental gradient between the shoulder and base of tomato fruit. *SlTALE2* and *SlTALE5* (i.e., *SlTKN4* and *SlTKN2*) have been identified to play a role in regulating chloroplast development and green shoulder formation in tomato fruit [22]. However, *SlTALE22*, known as *SlBEL11*, was not differentially expressed between the shoulder and base of tomato fruit, and it has been reported to be involved in the regulation of chloroplast development and chlorophyll accumulation in tomato fruit [27].

Complex interactions between these SlTALEs were detected based on protein–protein interaction networks. SlTALE5 and SlTALE24 could interact with SlTALE6, SlTALE9, SlTALE11, and SlTALE19, and SlTALE5 could interact with SlTALE20 and SlTALE21. This suggested a possible interaction among these *SlTALE* genes in tomato fruit development, which is consistent with the frequent interactions observed between BELL and KNOX members. The predicted interaction networks provide new insights into how the regulation of *TALEs* affects the growth and development of tomato fruit, including chloroplast development, chlorophyll accumulation, and the formation of green fruit shoulder.

Immature green fruits are capable of photosynthesis, and up to 20% of the total fruit carbohydrate comes from the photosynthesis of fruit chloroplasts in tomato [46]. During fruit maturation, fruit chloroplasts develop into chromoplasts to synthesize and accumulate nutrients and flavor compounds [47]. It is an effective way to improve fruit quality by increasing the abundance and function of chloroplasts. In this study, *GS* tomato fruits with dark green shoulders had significantly enhanced chloroplast development and chlorophyll accumulation, resulting in a significant increase in the level of photosynthesis-related parameters (chlorophyll fluorescence parameters) (Figure 5). As mentioned previously, four *SlTALE* genes had been reported to affect the development of tomato chloroplasts and fruit quality [17,27,28]. As positive regulators, *SlTKN2* and *SlTKN4* are promising candidates to target for improving fruit phytonutrients. Negative regulators, such as *SlBEL4* and *SlBEL11*, can also be the ideal targets to be precisely edited by CRISPR/Cas9 to create fruits with improved phytonutrients. Notably, the TALE family is a pivotal transcription factor and involves many processes of plant growth and development, including some core processes such as floral meristem formation, organ morphogenesis, and fruit development [8,48]. When *SlTALEs* are applied to tomato breeding programs, they may have negative pleiotropic effects that affect other aspects of plant growth. That is a strategy to specifically target *SlTALEs* in fruits by using fruit-specific promoters or RNAi technology to avoid the negative or pleiotropic effects on plant growth. This study suggested several candidate genes for further analysis to improve fruit quality by regulating tomato fruit chloroplast development.

## 4. Materials and Methods

### 4.1. Plant Materials

The tomato (*Solanum lycopersicum* L.) inbred line TTD302A and mutant line *GS* were used in this study. The mutant line *GS* (in which immature fruit possess a green shoulder) derived from ethyl methyl sulfonate (EMS) mutagenesis of inbred line TTD302A (in which immature fruit lack a green shoulder). All plant materials were grown in a glasshouse under a 16-h light/8-h dark photoperiod, with 60–75% relative humidity and ambient temperature of 20 °C (night) to 30 °C(day), at Northwest A&F University, Yangling, Shaanxi. Flowers were tagged at anthesis, and fruit development was recorded as days post-anthesis (dpa). The different stages of fruit development included mature green (MG) at 42 dpa, which was characterized by bright green coloration without obvious color change; the first sign of color change was observed at the breaker (Br) stage at approximately 45 dpa; the Br+5 stage occurred after 5 days of the Br stage and/or the pink ripe (P) stage. All plant samples were collected at the same time each day. The samples were directly frozen in liquid nitrogen and stored at −80 °C for further use.

### 4.2. Data Sources

Using the latest tomato reference genomes (SL4.0 and ITAG4.0), the tomato genomes, amino acids, CDS sequence assembly, and corresponding annotation were obtained from the Sol Genomics Network (https://solgenomics.net, accessed on 3 February 2021). *Solanum lycopersicum*, *Solanum tuberosum*, *Arabidopsis thaliana*, *Hordeum vulgare*, *Nicotiana benthamiana*, *Oryza rufipogon*, and *Vitis vinifera* TALE family protein sequences were downloaded from the PlantTFDB database [49] (http://planttfdb.cbi.pku.edu.cn, accessed on 6 November 2021) (Fasta S1 and Appendix A).

### 4.3. Identification of SlTALE Family Members

To identify the TALE family members of tomato, previously reported *Arabidopsis* and tomato TALE proteins sequences were used as queries to search against the tomato protein database with the BLASTP program (https://blast.ncbi.nlm.nih.gov, accessed on 3 November 2021) using an E-value below 10^−3^ and an identity of 50% as the threshold. In addition, the Hidden Markov Model constructed by the HMMER 3.0 program was used to search the tomato protein sequences to identify TALE family members. Redundant sequences obtained from the HMMER and BLAST results were removed following the alignment of the predicted SlTALE members. The predicted SlTALE members were then aligned for comparison and removal of redundancy. The aligned sequences were used as candidate SlTALE family sequences. All candidate sequences were confirmed to be SlTALE proteins using PfamScan (https://www.ebi.ac.uk/Tools/pfa/pfamscan/, accessed on 15 November 2021) and SMART (http://smart.embl-heidelberg.de/, accessed on 15 November 2021).

### 4.4. Sequence Analysis of SlTALE Family Members

The physical and chemical properties of SlTALE proteins were predicted with ExPASy (https://web.expasy.org/protparam/, accessed on 2 December 2021). The signal peptide of SlTALE proteins was determined by the SignalP 5.0 Server (http://www.cbs.dtu.dk/services/SignalP, accessed on 2 December 2021). Subcellular localization of SlTALE proteins was performed using Cell-PLoc (http://www.csbio.sjtu.edu.cn/bioinf/Cell-PLoc−2/, accessed on 2 December 2021) and WoLF PSORT (http://www.genscript.com/wolf-psort.html, accessed on 2 December 2021).

### 4.5. Gene Structure, Conserved Motif, and Phylogenetic Analysis

Conserved motifs of SlTALE family proteins were identified by MEME software (http://meme-suite.org/, accessed on 7 December 2021), with motif width of 6–50 and maximum motif number of 12. The gene structures and chromosomal locations of *SlTALE* family genes were characterized by comparing the coding and genomic sequences with TBtools [50]. Gene duplication information was obtained by aligning all SlTALE family genes.

Multiple sequence alignments of candidate TALE family proteins from *Solanum lycopersicum*, *Solanum tuberosum*, *Arabidopsis thaliana*, *Hordeum vulgare*, *Nicotiana benthamiana*, *Oryza rufipogon*, and *Vitis vinifera* were performed using MAFFT. The phylogenetic tree was constructed with MEGA6 software using the neighbor-joining (NJ) method and 1000 bootstrap replicates, and the output was visualized using the online software tool EvolView (http://www.evolgenius.info/, accessed on 7 December 2021) and FigTree (v1.4.4) software. Finally, a figure displaying the phylogenetic tree, conserved motifs, and gene structure was built using TBtools.

### 4.6. Cis-elements, TF Binding Sites, and Protein Interaction Network

To analyze the cis-elements and TF binding sites of the promoters, the 2000-bp region upstream from the start codon was obtained from the tomato genome sequence by a Perl script (Perl S1) and TBtools. Cis-elements were identified by PlantCARE (http://bioinformatics.psb.ugent.be/webtools/plantcare/html/, accessed on 29 November 2021). TF binding sites were searched by Plant Transcriptional Regulatory Map (http://plantregmap.gao-lab.org/binding_site_prediction.php, accessed on 5 December 2021) [51,52]. The protein-protein interaction network of the SlTALE family was analyzed by String (https://string-db.org/, accessed on 7 December 2021) [53].

### 4.7. Chlorophyll Content, Chlorophyll Fluorescence Parameters, and Chloroplast Structure

The chlorophyll content was determined spectrophotometrically following a previously described method [54]. Fruit samples were collected at 28 dpa, and the chlorophyll content was measured using the stem end (fruit shoulder area) and base (fruit navel area) parts of the fruits.

After the fruit at 28 dpa was conditioned in the dark for 30 min, a Multispectral Fluorescence Imaging System (Fc800 FluorCam, Photon Systems Instruments, Drásov, Czech Republic) was used for chlorophyll fluorescence dynamic imaging and measurements of chlorophyll fluorescence parameters.

The pericarp was isolated and fixed in 0.1 M cacodylate buffer consisting of 4% glutaraldehyde at 4 °C for 24 h, postfixed in 1% osmium tetroxide, and dehydrated in a gradient ethanol series. After embedding the sample in Spurr resin, the ultrathin sections of samples were obtained using the Leica UC7 ultramicrotome (Leica, Solms, Germany) on 200 mesh copper grids, and then stained with uranyl acetate and lead citrate. The Hitachi HT7700 transmission electron microscope (Hitachi Ltd., Tokyo, Japan) was used to observe the ultrastructures of plastids.

### 4.8. Transcriptome Analysis and Expression Analysis of SlTALE Family Genes

The pericarp of the shoulder region from WT and *GS* tomato fruits was taken for RNA-seq analysis at five critical periods (Appendix A), including 7, 14, 21, and 28 dpa as well as at P (Pink ripening period, or Br+5). Three biological replicates were collected from each tissue type and three technical replicates within each biological replicate. Library construction was performed using the Illumina TruseqTM RNA Sample Prep Kit, and RNA-seq was performed on the Illumina NovaSeq 6000 sequencing platform (Illumina, Shanghai, China) by Shanghai Majorbio Bio-pharm Technology Co., Ltd The data were analyzed using the Majorbio Cloud Platform (www.majorbio.com, accessed on 18 September 2021). In addition, RNA-seq data (Appendix A) for “Ailsa Craig” tomato (which has a dark green shoulder) were downloaded from the TOMExpress website (http://tomexpress.toulouse.inra.fr, accessed on 28 September 2021) at different developmental stages (10 dpa, 20 dpa, MG, Br, and Br+5) between the shoulder and the base of the fruit, to investigate the expression profile of *SlTALE* family genes in different tomato materials.

A total of 16 differentially expressed or highly expressed genes were selected for relative expression analysis to validate the RNA-seq results. For quantitative RT-PCR, total RNA was extracted using an RNA Extraction Kit (Omega Bio-Tek, Shanghai, China), followed by an RNA integrity assay with 1% agarose gels; the quality and concentration of RNA were assessed using a Nanodrop 2000 spectrophotometer (Thermo Scientific, Waltham, MA, USA). First-strand cDNA was reverse-transcribed using an *Evo M-MLV* Mix Kit with gDNA Clean for qPCR (Accurate Biotechnology, Changsha, China). qRT-PCR was performed using gene-specific primers with the SYBR^®^ Green Premix Pro Taq HS qPCR Kit (Accurate Biotechnology, Changsha, China) in a QuantStudio^®^5 Real-Time PCR System (Life Technologies, Carlsbad, CA, USA). Three biological replicates were collected from each tissue type and three technical replicates within each biological replicate. The *SlActin* (*Solyc01g104770*) was used as the reference gene. The relative expression levels of the 16 *SlTALE* genes were calculated according to the 2^−ΔΔCt^ method. Primers used for qRT-PCR are shown in Appendix A.

## 5. Conclusions

In this study, a total of 24 *SlTALE* genes were identified. The physicochemical characteristics, subcellular localization, phylogenetic analysis, gene structure, motifs, promoter activity, and expression profiles were investigated with diverse bioinformatics methods. Phylogenetic analysis showed that 24 SlTALE members could be classified into 5 BELL subfamilies and 4 KNOX subfamilies, and the members of the same subfamily had similar gene structures and motifs. The cis-elements responsive to light, phytohormone, developmental regulation, and environmental stress were enriched in the promoter of *SlTALE* genes, and the light response elements were the most abundant. An abundance of TF binding sites was also enriched in the promoter of *SlTALE* genes. An analysis of gene expression profiles indicated that six *SlTALE* genes (*SlTALE2*, *SlTALE3*, *SlTALE5*, *SlTALE11*, *SlTALE20*, and *SlTALE21*) were highly expressed in the *GS* fruit shoulder region (Gs), and four *SlTALE* genes (*SlTALE6*, *SlTALE9*, *SlTALE19*, and *SlTALE24*) were highly expressed in the parts with less-developed chloroplasts, such as Gb, Ws, and Wb. The protein-protein interaction networks predicted that a complex regulatory network controls the expression of these *SlTALE* genes during chloroplast development and green fruit shoulder formation. Overall, our findings provide new insight into *SlTALE* genes in tomato and their role in fruit chloroplast development, and suggested candidate genes for future research aimed at improving fruit quality by regulating tomato fruit chloroplast development.

## Figures and Tables

**Figure 1 ijms-23-04507-f001:**
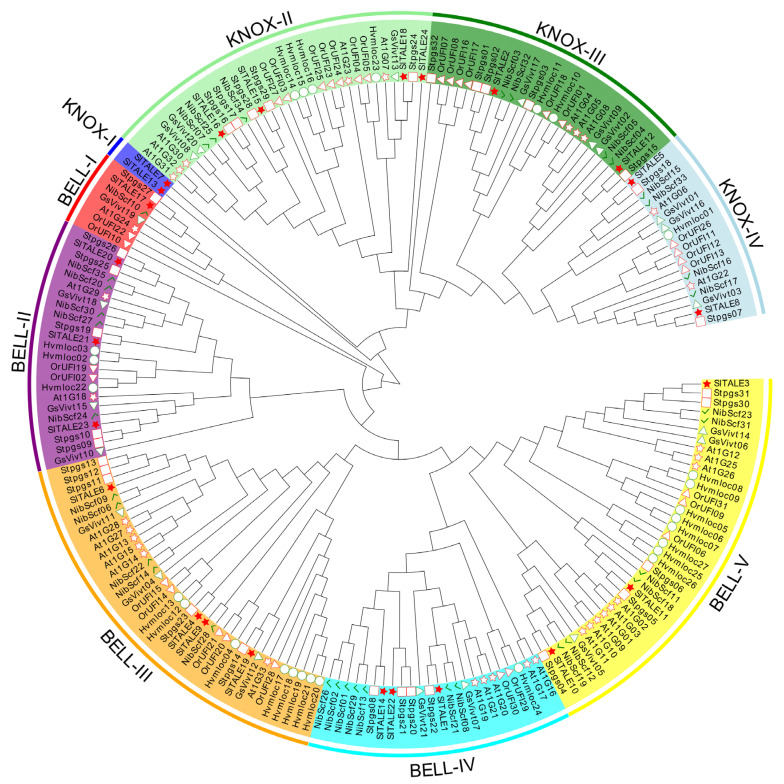
Phylogenetic tree of the TALE family. Phylogenetic relationships among 203 TALE proteins in *Solanum lycopersicum*, *Solanum tuberosum*, *Arabidopsis thaliana*, *Hordeum vulgare*, *Nicotiana benthamiana*, *Oryza rufipogon*, and *Vitis vinifera*. The phylogenetic tree was constructed using MEGA6 with a neighbor-joining method and 1000 bootstrap replicates. *Solanum lycopersicum*, *Solanum tuberosum*, *Arabidopsis thaliana*, *Hordeum vulgare*, *Nicotiana benthamiana*, *Oryza rufipogon*, and *Vitis vinifera* were marked with a red star, red rectangle, white red star, white green circle, green check, white red triangle, and white green triangle, respectively.

**Figure 2 ijms-23-04507-f002:**
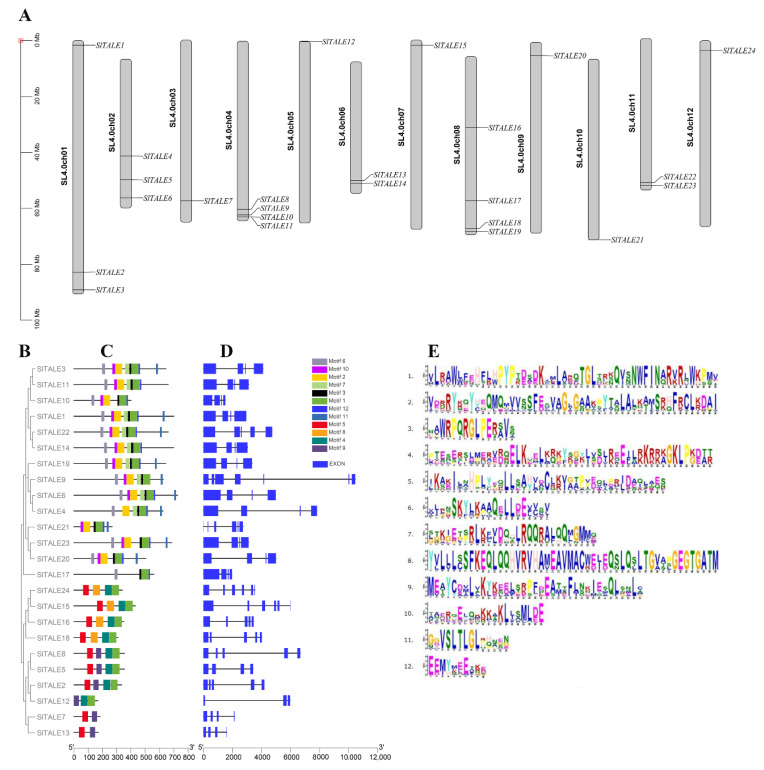
Chromosomal locations, phylogenetic tree, conserved motifs, and gene structures of SlTALE family members. (**A**) Chromosomal locations of *SlTALE* genes. (**B**) The phylogenetic tree of the SlTALE family. (**C**) Conserved motifs in SlTALE proteins. The motifs were identified by the MEME Suite. Twelve different conserved motifs were displayed in different colored boxes. (**D**) Gene structures of the *SlTALE* genes. The intron/exon structure was mapped by TBtools, and introns are indicated by black lines. (**E**) Sequence logos of 12 conserved motifs were identified.

**Figure 3 ijms-23-04507-f003:**
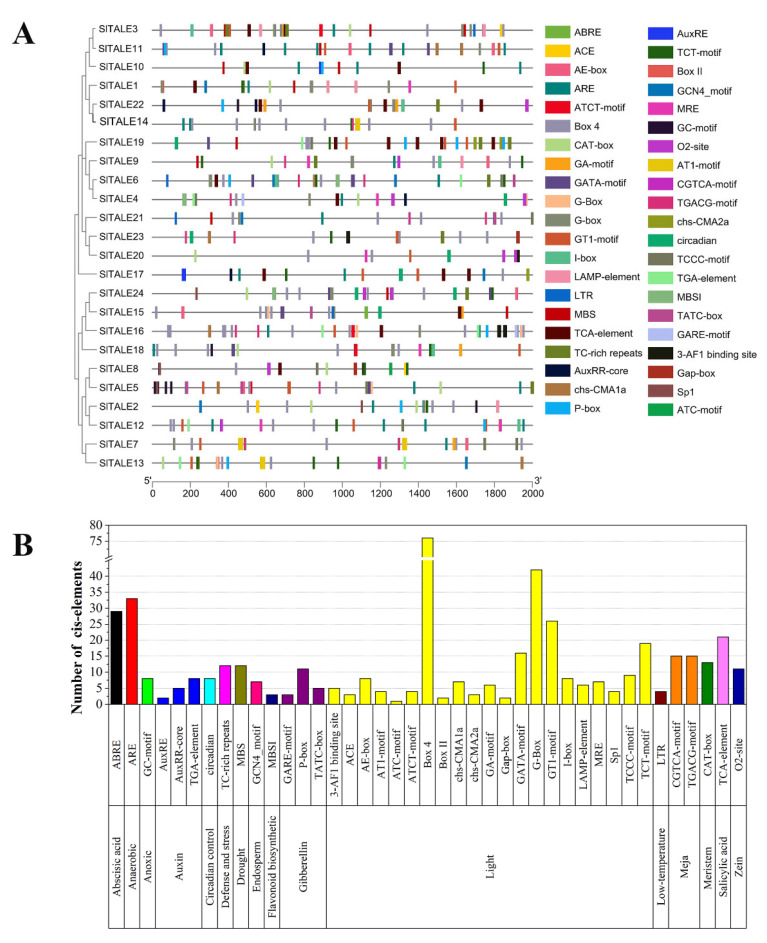
Analysis of the cis-elements and TF binding sites of *SlTALE* genes. (**A**) Cis-element analysis of *SlTALE* genes. (**B**) The number of the cis-elements of *SlTALE* genes. (**C**) TF binding sites analysis of *SlTALE* genes.

**Figure 4 ijms-23-04507-f004:**
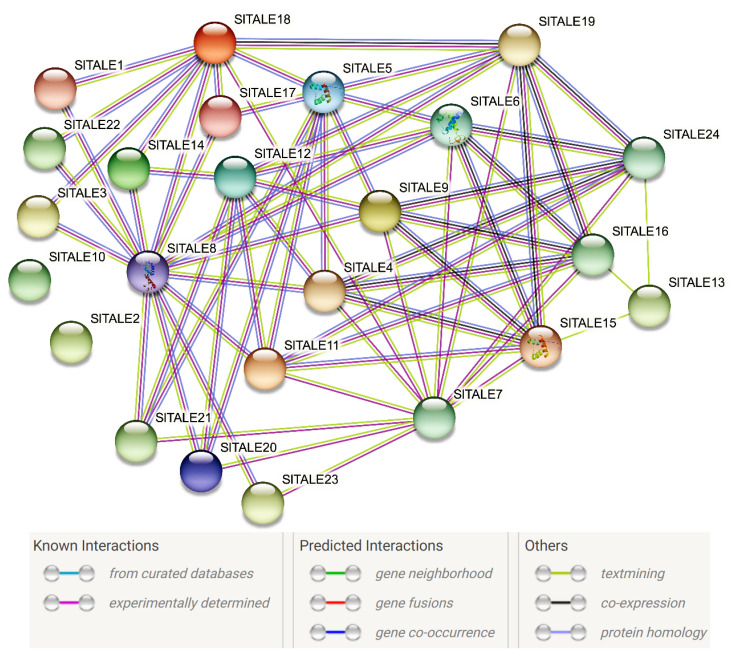
Predicted functional interaction networks of SlTALE proteins in *Solanum lycopersicum*.

**Figure 5 ijms-23-04507-f005:**
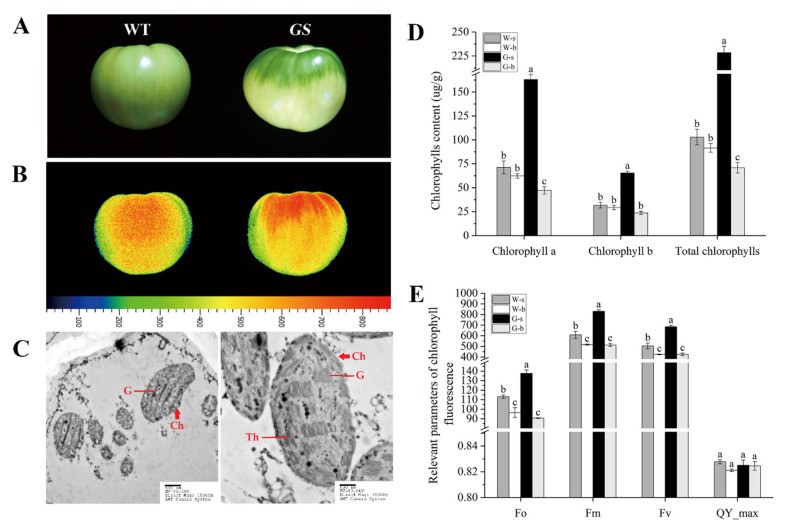
Fruit phenotype and chloroplast development in the *GS* mutant and WT (Inbred line TTD302A). (**A**) Fruit phenotype at 28 dpa. (**B**) The chlorophyll fluorescence dynamic imaging of fruit. (**C**) Chloroplast ultrastructure (bars, 0.5 µm) of the fruit shoulder pericarp at 28 dpa was observed by transmission electron microscope. (**D**) Chlorophyll content of the pericarp at 28 dpa. (**E**) Chlorophyll fluorescence parameters of the fruit shoulder pericarp at 28 dpa. W-s and W-b indicate the shoulder and base of WT fruit, respectively. G-s and G-b indicate the shoulder and base of the *GS* mutant fruit, respectively. Ch, chloroplast. G, grana. Th, thylakoid. Fo, minimal fluorescence. Fm, maximal fluorescence. Fv, variable fluorescence. QY_max, maximal quantum yield of PSⅡ. The values shown are the means ± standard deviation of three replicates. The different letters in a column denote significant differences among the treatments at *p* < 0.05.

**Figure 6 ijms-23-04507-f006:**
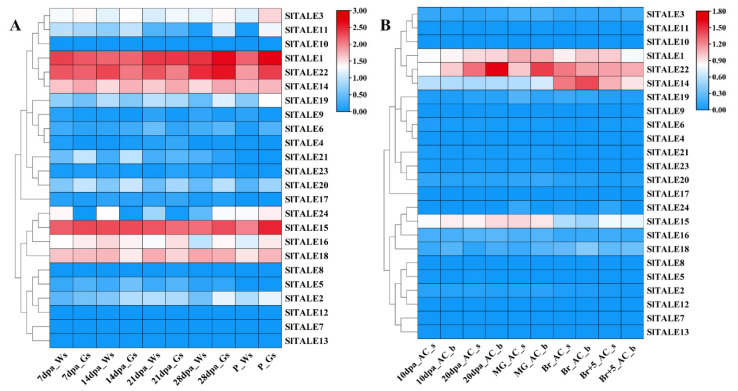
Expression patterns of *SlTALE* genes during chloroplast development and green shoulder formation in tomato fruit. (**A**) Expression patterns of *SlTALE* genes in the shoulder pericarp of WT and *GS* fruits. (**B**) Expression patterns of *SlTALE* genes in the shoulder and base pericarp of AC (Ailsa Craig) fruit. The RNA-seq data were normalized by FPKM (fragments per kilobase of exon per million fragments mapped) to construct the heatmap by TBtools software. The dpa indicates days post-anthesis, and Br+5 means 5 days after the breaker stage. W-s and W-b indicate the shoulder and base of WT fruit, respectively. G-s and G-b indicate the shoulder and base of the *GS* mutant fruit, respectively. The colored scale varies from blue to yellow, indicating relatively low or high expression, respectively.

**Figure 7 ijms-23-04507-f007:**
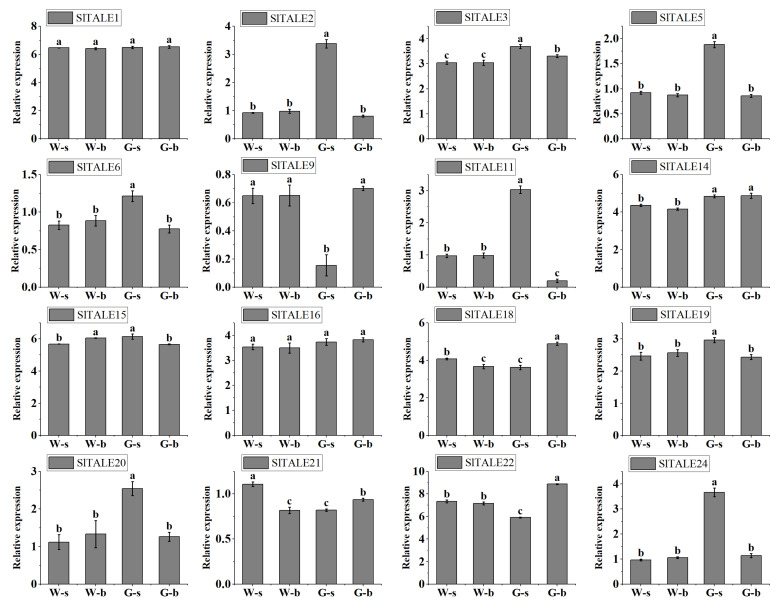
Confirmation of the expression patterns of *SlTALE* genes in fruit by qRT–PCR. Sixteen differentially expressed or highly expressed genes were used, as well as the pericarp of the tomato fruit at 28 dpa. The values shown are the means ± standard deviation of three replicates. Different letters among columns denote significant differences among the treatments at *p* < 0.05.

## Data Availability

All data are provided within the main text and Appendix A.

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
