# Peer review of "Identification of TALE Transcription Factor Family and Expression Patterns Related to Fruit Chloroplast Development in Tomato (Solanum lycopersicum L.)"

_ijms, 2022, doi:10.3390/ijms23094507_

Round 1
Reviewer 1 Report
Research Article: “Identification of TALE Transcription Factor Family and Expression Patterns Related to Fruit Chloroplast Development in Tomato (Solanum lycopersicum L.)”
In this manuscript authors have identified 24 genes of the TALE family in tomato and classified into five BELL subfamily and four KNOX subfamily. According to their promoter activities analysis disclosed that SlTALE genes were involved in light responsiveness regulation, phytohormone response, developmental regulation and controlled by an array of transcription factors. They uncovered that the SlTALE genes in tomato and their involvement in fruit chloroplast development could stipulate an academic basis for the functional research and application of the TALE gene family. This study is a good read however, this reviewer has few comments that can improve the quality of the manuscript.
1, Authors can add one paragraph for abbreviations.
2, The English of manuscript needs to be polished (minor) and there are few typo errors in the manuscript that can be checked.
3, At least one illustrative figure may be provided as to highlight the summary of this study.
4, All figures quality may be improved (high resolution).
5, Abstract and conclusion are the same in this manuscript, better to rewrite the conclusions.
6, Did authors find any link between ethylene/RIN role in SlTALE genes regulation, explain.
7, Do SlTALE genes only involve in fruit chloroplast development? Or they play in major role in whole plant chloroplast development. Add discussion.
Author Response
Dear Editors and Reviewers:
Thank you for your letter and for the reviewers’ comments concerning our manuscript entitled “Identification of TALE Transcription Factor Family and Expression Patterns Related to Fruit Chloroplast Development in Tomato (Solanum lycopersicum L.)” (ID: ijms-1659860). Those comments are all valuable and very helpful for revising and improving our paper, as well as the important guiding significance to our researches. We have studied comments carefully and have made correction which we hope meet with approval. Revised portion are marked in red in the paper. The main corrections in the paper and the responds to the reviewer’s comments are as flowing:
Reviewer #1:
Point 1: Authors can add one paragraph for abbreviations.
Response 1: One paragraph for abbreviations was added according to the Reviewer’s suggestion.
Point 2: The English of manuscript needs to be polished (minor) and there are few typo errors in the manuscript that can be checked.
Response 2: Manuscript was further edited by a native English speaker according to the Reviewer’s comments.
Point 3: At least one illustrative figure may be provided as to highlight the summary of this study.
Response 3: A illustrative figure was added according to the Reviewer’s comments. It was provided within supplementary materials.
Point 4: All figures quality may be improved (high resolution).
Response 4: All figures' quality had been improved according to the Reviewer’s comments.
Point 5: Abstract and conclusion are the same in this manuscript, better to rewrite the conclusions.
Response 5: The conclusions were rewritten according to the Reviewer’s comments.
Point 6: Did authors find any link between ethylene/RIN role in SlTALE genes regulation, explain.
Response 6: In this study, no link was found between the role of ethylene/RIN and the regulation of SlTALE genes. On the one hand, no ethylene-responsive element was found in the promoter region of the SlTALE gene (Fig 3B), indicating that the expression of SlTALE genes was not regulated by the ethylene signaling pathway. On the other hand, the stage of chloroplast development in tomato fruit was mainly before fruit ripening, while ethylene production and action were mainly during fruit ripening, and they might be independent of each other in time or developmental stage.
Point 7: Do SlTALE genes only involve in fruit chloroplast development? Or they play in major role in whole plant chloroplast development. Add discussion.
Response 7: Not only, SlTALE genes are involved in many processes of plant growth and development, including some core processes such as floral meristem formation, organ morphogenesis and fruit development. It was added in discussion according to the Reviewer’s comments.
We tried our best to improve the manuscript and made some changes in the manuscript. These changes will not influence the content and framework of the paper. And here we did not list the changes but marked in red in revised paper.
We appreciate for Editors/Reviewers’ warm work earnestly, and hope that the correction will meet with approval.
Once again, thank you very much for your comments and suggestions.

Reviewer 2 Report
Review of ijms-1659860
Identification of TALE Transcription Factor Family and Expression Patterns Related to Fruit Chloroplast Development in Tomato (Solanum lycopersicum L.)
Jin Wang, Pan Zhao, Baohui Cheng, Yanhong Zhang, Yuanbo Shen, Xinyu Wang, Qinghua Zhang, Qianqi Lou, Shijie Zhang, Bo Wang, Shiming Qi, Yushun Li, MD.Monirul Islam, Tayeb Muhammad, Fei Zhang and Yan Liang
The authors set out to characterize all the tomato genes encoding TALE transcription factors. They first HMMER and BLASTP to query the tomato genome, and then verified putative candidates using PfamScan and SMART, identifying a total of 24 SITALE genes. They next constructed a phylogenetic tree of TALE plant proteins by aligning them with TALE proteins from Solanum tuberosum, Arabidopsis thaliana, Hordeum vulgare, Nicotiana benthamiana, Oryza rufipogon and Vitis vinifera. They found that the tomato proteins could be subdivided into 5 BELL subfamilies and 4 KNOX subfamilies. Next they determined the chromosomal locations and gene structures of each of their candidate genes, and a detailed analysis of all the tomato TALE proteins including the length of their coding sequence and of the proteins, their predicted molecular weights, PIs and hydrophobicities. They also identified conserved domains and motifs, and examined their promoter regions 5’ to the start for conserved cis-acting elements. They next used the STRING application to predict interaction networks, and identified a number of potential interactions. They concluded by comparing their expression in the tomato green shoulder mutant and the wild type from which it was derived. They first compared chlorophyll content and various light-harvesting parameters in the tops and bottoms of wild type and mutant fruits, and showed that the mutant had more chlorophyll and more light harvesting activity in the top and less in the bottom than wild type. Next they performed RNAseq on these tissues, and identified a few SITALE genes that appeared to be differentially expressed in the shoulders of mutant and wild type plants. They therefore chose to validate these results by RTR-qPCR, and identified several SITALE genes that were significantly upregulated in the shoulders of the mutant fruits and one that was significantly down-regulated compared to wild type.
Overall, the study seems to have been competently performed using suitable techniques, adequate replication and appropriate statistical analysis (although some of this needs clarification, as noted below). I therefore think that it is worth sharing with the plant community after correcting the issues noted below.
My biggest concern is: how does this study improve our knowledge of plant development, or even improve tomato breeding? Yes, the authors have provided a detailed analysis of the tomato TALE transcription factor family, but how can this information be applied? Can they suggest any candidate genes for further analysis, eg by overexpression of CRISPR/Cas9 knockout? What does their analysis of SITALE expression in the green shoulder mutant tell us about why this mutant has this phenotype and what it tells us about fruit development? How can their study be used to breed better tomatoes?
The abstract needs to summarize the paper better. For example, it should provide more information about the green shoulder mutant and the physiological parameters shown in figure 5. It should also provide more information about their other key findings.
I am also concerned about the over-reliance on protein interactions based on computational analysis. In figure 4 I find it very hard to determine which interactions have been determined experimentally, which in my opinion are much more believable, and which have been inferred. A thick black line linking interactions that had been validated experimentally with thin lines of the various colors indicated inferred interactions and how they were inferred would make figure 4 much more believable.
Section 4.8 should indicate how many biological replicates were collected from each tissue type, and how many technical replicates were performed on each biological replicate.
Line 207: caption to figure 5 should indicate the number of samples, whether the error bars represent St. deviation or St. error, what the different letters mean and how the statistical significance was determined.
Line 242: I do not see any data presented for either W-b or G-b in figure 6-a, nor do I see any mention of RNAseq data from the base in the text. Are these needed?
Line 259: caption to figure 7 should indicate whether these represent three biological replicates, or three technical replicates (there should be at least 3 biological replicates for each tissue type, and three technical replicates performed on each biological replicate). This caption should also state the reference gene used, and the method used to determine the relative expression levels.
It would be useful to indicate which SITALES had been previously studied, and what they were called in these studies. Perhaps this could be indicated in Table 1
The second section 2.6 (two sections are denoted 2.6) needs to be rewritten to highlight the significant differences. As written, they get lost.
Overall the English is quite good, but there are numerous mistakes and some make the meaning hard to understand. I therefore recommend editing by a native English speaker.
Throughout the paper “BELL subfamily” should be “BELL subfamilies” and “KNOX subfamily” should be “KNOX subfamilies.
Throughout the paper please clarify what you mean by “light responsiveness regulatory.” This doesn’t make sense by itself. Is this a category based on promoter element analysis? Similarly, “phytohormone responsive,”, ‘developmentally regulated “ and “environmental stress-responsive” are also awkward but not as bad.
Line 15: you should explain that the green shoulder (GS) mutant shows uneven pigment distribution, and that the distribution of these SITALE mRNAs correlates with pigmentation.
Lines 21-31 are hard to understand and should be rewritten for clarity.
Lines 37-39 are hard to understand and should be rewritten for clarity.
Lines 64-70 are hard to understand and should be rewritten for clarity.
Line 88: how were the sequences compared?
Lines 92-95 are hard to understand and should be rewritten for clarity.
Lines 101-103 are hard to understand and should be rewritten for clarity.
Lines 125-128 are hard to understand and should be rewritten for clarity.
Line 129 should clarify they they have the most SITALE genes.
Lines 130-134 are hard to understand and should be rewritten for clarity.
Lines 136-139 are hard to understand and should be rewritten for clarity.
Lines 152-163 are hard to understand and should be rewritten for clarity.
Lines 168-169 are hard to understand and should be rewritten for clarity.
Lines 188-190 are hard to understand and should be rewritten for clarity.
Line 235 “conservative” should be “conserved”
Lines 261-264 are hard to understand and should be rewritten for clarity.
Lines 319-322 are hard to understand and should be rewritten for clarity.
Line 330 “mutagenization” should be “mutagenesis”
Lines 355-356 are hard to understand and should be rewritten for clarity.
Line 361: How did you use PfamScan to confirm that these were SITALE proteins? What criteria did you use? Cut-offs?
Lines 364-366 are hard to understand and should be rewritten for clarity.
Lines 424-427 are hard to understand and should be rewritten for clarity.
Author Response
Dear Editors and Reviewers:
Thank you for your letter and for the reviewers’ comments concerning our manuscript entitled “Identification of TALE Transcription Factor Family and Expression Patterns Related to Fruit Chloroplast Development in Tomato (Solanum lycopersicum L.)” (ID: ijms-1659860). Those comments are all valuable and very helpful for revising and improving our paper, as well as the important guiding significance to our researches. We have studied comments carefully and have made correction which we hope meet with approval. Revised portion are marked in red in the paper. The main corrections in the paper and the responds to the reviewer’s comments are as flowing:
Reviewer #2:
Point 1: My biggest concern is: how does this study improve our knowledge of plant development, or even improve tomato breeding? Yes, the authors have provided a detailed analysis of the tomato TALE transcription factor family, but how can this information be applied? Can they suggest any candidate genes for further analysis, eg by overexpression of CRISPR/Cas9 knockout? What does their analysis of SITALE expression in the green shoulder mutant tell us about why this mutant has this phenotype and what it tells us about fruit development? How can their study be used to breed better tomatoes?
Response 1: According to the Reviewer’s comments, we made some discussions about the regulation of chloroplast development by TALE family genes, the effect of chloroplast function on tomato fruit quality, and the application of TALE family genes in tomato quality improvement and breeding. The details were presented in the last paragraph of the discussion.
Point 2: The abstract needs to summarize the paper better. For example, it should provide more information about the green shoulder mutant and the physiological parameters shown in figure 5. It should also provide more information about their other key findings.
Response 2: The abstract was rewritten according to the Reviewer’s comments.
Point 3: I am also concerned about the over-reliance on protein interactions based on computational analysis. In figure 4 I find it very hard to determine which interactions have been determined experimentally, which in my opinion are much more believable, and which have been inferred. A thick black line linking interactions that had been validated experimentally with thin lines of the various colors indicated inferred interactions and how they were inferred would make figure 4 much more believable.
Response 3: According to the Reviewer’s comments, we improved the figures' quality to make the protein interactions links clearer and more explicit in Figure 4. A thick purple line linking interactions that had been validated experimentally in Figure 4, and thin lines of the other various colors indicated inferred interactions.
Point 4: Section 4.8 should indicate how many biological replicates were collected from each tissue type, and how many technical replicates were performed on each biological replicate.
Response 4: Three biological replicates were collected from each tissue type and 3 technical replicates within each biological replicate. It was added according to the Reviewer’s comments.
Point 5: Line 207: caption to figure 5 should indicate the number of samples, whether the error bars represent St. deviation or St. error, what the different letters mean and how the statistical significance was determined.
Response 5: The values shown are the means ± standard deviation of three replicates. The different letters in a column denote significant differences among the treatments at p < 0.05. It was added according to the Reviewer’s comments.
Point 6: Line 242: I do not see any data presented for either W-b or G-b in figure 6-a, nor do I see any mention of RNAseq data from the base in the text. Are these needed?
Response 6: The RNA-seq data of W-b or G-b about SITALES are largely similar to W-s, as they both belong to fruiting regions with underdeveloped chloroplasts, and this is clearly reflected r in the qRT-PCR data (Fig. 7). Therefore, the RNA-seq data of W-b or G-b are not essential.
Point 7: Line 259: caption to figure 7 should indicate whether these represent three biological replicates, or three technical replicates (there should be at least 3 biological replicates for each tissue type, and three technical replicates performed on each biological replicate). This caption should also state the reference gene used, and the method used to determine the relative expression levels.
Response 7: Three biological replicates were collected from each tissue type and 3 technical replicates within each biological replicate. The SlActin (Solyc01g104770) was used as a reference gene. The relative expression levels of the 16 SlTALE genes were calculated according to the 2−ΔΔCt method. These are mentioned in the method and some information was also added according to the Reviewer’s comments.
Point 8: It would be useful to indicate which SITALES had been previously studied, and what they were called in these studies. Perhaps this could be indicated in Table 1
Response 8: The name and related references of SITALEs that had been previously studied were indicated in Table 1 according to the Reviewer’s comments.
Point 9: The second section 2.6 (two sections are denoted 2.6) needs to be rewritten to highlight the significant differences. As written, they get lost.
Response 9: The second section 2.6 (should be 2.7) had been rewritten according to the Reviewer’s comments.
Point 10: Overall the English is quite good, but there are numerous mistakes and some make the meaning hard to understand. I therefore recommend editing by a native English speaker.
Response 10: Manuscript was further edited by a native English speaker according to the Reviewer’s comments.
Point 11: Throughout the paper “BELL subfamily” should be “BELL subfamilies” and “KNOX subfamily” should be “KNOX subfamilies.
Response 11: The“BELL subfamily”and “KNOX subfamily” had been corrected to “BELL subfamilies” and ““KNOX subfamilies” according to the Reviewer’s comments.
Point 12: Throughout the paper please clarify what you mean by “light responsiveness regulatory.” This doesn’t make sense by itself. Is this a category based on promoter element analysis? Similarly, “phytohormone responsive”, ‘developmentally regulated“ and “environmental stress-responsive” are also awkward but not as bad.
Response 12: According to the Reviewer’s comments, “light responsiveness regulatory”, “phytohormone responsive”, “developmentally regulated“ and “environmental stress-responsive” had been corrected to “light”, “phytohormone”, “developmental regulation “ and “environmental stress” , respectively.
Point 13: Line 15: you should explain that the green shoulder (GS) mutant shows uneven pigment distribution, and that the distribution of these SITALE mRNAs correlates with pigmentation.
Response 13: According to the Reviewer’s comments, it was mentioned in the abstract that had been rewritten.
Point 14: Line 88: how were the sequences compared?
Response 14: We are very sorry for our incorrect writing, the correct writing should be “The two identification results were combined and taken together”
Point 15: Line 361: How did you use PfamScan to confirm that these were SITALE proteins? What criteria did you use? Cut-offs?
Response 15: The BELL domains (POX, homeodomain) or KNOX domains (KNOX, homeodomain) were TALE gene family conserved. The SITALE proteins’ domains were searched by PfamScan to determine whether it has BELL domains or KNOX domains.
Point 16: Other comments:
Lines 21-31 are hard to understand and should be rewritten for clarity.
Lines 37-39 are hard to understand and should be rewritten for clarity.
Lines 64-70 are hard to understand and should be rewritten for clarity.
Lines 92-95 are hard to understand and should be rewritten for clarity.
Lines 101-103 are hard to understand and should be rewritten for clarity.
Lines 125-128 are hard to understand and should be rewritten for clarity.
Line 129 should clarify they they have the most SITALE genes.
Lines 130-134 are hard to understand and should be rewritten for clarity.
Lines 136-139 are hard to understand and should be rewritten for clarity.
Lines 152-163 are hard to understand and should be rewritten for clarity.
Lines 168-169 are hard to understand and should be rewritten for clarity.
Lines 188-190 are hard to understand and should be rewritten for clarity.
Line 235 “conservative” should be “conserved”
Lines 261-264 are hard to understand and should be rewritten for clarity.
Lines 319-322 are hard to understand and should be rewritten for clarity.
Line 330 “mutagenization” should be “mutagenesis”
Lines 355-356 are hard to understand and should be rewritten for clarity.
Lines 364-366 are hard to understand and should be rewritten for clarity.
Lines 424-427 are hard to understand and should be rewritten for clarity.
Response 16: According to the Reviewer’s comments, we have rewritten or revised these sections. Details are presented in the manuscript marked in red.
We tried our best to improve the manuscript and made some changes in the manuscript. These changes will not influence the content and framework of the paper. And here we did not list the changes but marked in red in revised paper.
We appreciate for Editors/Reviewers’ warm work earnestly, and hope that the correction will meet with approval.
Once again, thank you very much for your comments and suggestions.

Round 2
Reviewer 2 Report
The authors have satisfied all of my concerns, so I recommend publication.